# Graph Neural Tangent Kernel: Fusing Graph Neural Networks with Graph Kernels

**Simon S. Du**
Institute for Advanced Study
ssdu@ias.edu

**Kangcheng Hou**
Zhejiang University
kangchenghou@gmail.com

**Barnabás Póczos**
Carnegie Mellon University
bapoczos@cs.cmu.edu

**Ruslan Salakhutdinov**
Carnegie Mellon University
rsalakhu@cs.cmu.edu

**Ruosong Wang**
Carnegie Mellon University.
ruosongw@andrew.cmu.edu

**Keyulu Xu**
Massachusetts Institute of Technology
keyulu@mit.edu

## Abstract

While graph kernels (GKs) are easy to train and enjoy provable theoretical guarantees, their practical performances are limited by their expressive power, as the kernel function often depends on hand-crafted combinatorial features of graphs. Compared to graph kernels, graph neural networks (GNNs) usually achieve better practical performance, as GNNs use multi-layer architectures and non-linear activation functions to extract high-order information of graphs as features. However, due to the large number of hyper-parameters and the non-convex nature of the training procedure, GNNs are harder to train. Theoretical guarantees of GNNs are also not well-understood. Furthermore, the expressive power of GNNs scales with the number of parameters, and thus it is hard to exploit the full power of GNNs when computing resources are limited. The current paper presents a new class of graph kernels, Graph Neural Tangent Kernels (GNTKs), which correspond to *infinitely wide* multi-layer GNNs trained by gradient descent. GNTKs enjoy the full expressive power of GNNs and inherit advantages of GKs. Theoretically, we show GNTKs provably learn a class of smooth functions on graphs. Empirically, we test GNTKs on graph classification datasets and show they achieve strong performance.

## 1 Introduction

Learning on graph-structured data such as social networks and biological networks requires one to design methods that effectively exploit the structure of graphs. Graph Kernels (GKs) and Graph Neural Networks (GNNs) are two major classes of methods for learning on graph-structured data. GKs, explicitly or implicitly, build feature vectors based on combinatorial properties of input graphs. Popular choices of GKs include Weisfeiler-Lehman subtree kernel [Shervashidze et al., 2011], graphlet kernel [Shervashidze et al., 2009] and random walk kernel [Vishwanathan et al., 2010, Gärtner et al., 2003]. GKs inherit all benefits of kernel methods. GKs are easy to train, since the corresponding optimization problem is convex. Moreover, the kernel function often has explicit expressions, and thus we can analyze their theoretical guarantees using tools in learning theory. The downside of GKs, however, is that hand-crafted features may not be powerful enough to capture high-order

information that involves complex interaction between nodes, which could lead to worse practical performance than GNNs.

GNNs, on the other hand, do not require explicitly hand-crafted feature maps. Similar to convolutional neural networks (CNNs) which are widely applied in computer vision, GNNs use multilayer structures and convolutional operations to aggregate local information of nodes, together with non-linear activation functions to extract features from graphs. Various architectures have been proposed [Xu et al., 2019a, 2018]. GNNs extract higher-order information of graphs, which lead to more powerful features compared to hand-crafted combinatorial features used by GKs. As a result, GNNs have achieved state-of-the-art performance on a large number of tasks on graph-structured data. Nevertheless, there are also disadvantages of using GNNs. The objective function of GNNs is highly non-convex, and thus it requires careful hyper-parameter tuning to stabilize the training procedure. Meanwhile, due to the non-convex nature of the training procedure, it is also hard to analyze the learned GNNs directly. For example, one may ask whether GNNs can provably learn certain class of functions. This question seems hard to answer given our limited theoretical understanding of GNNs. Another disadvantage of GNNs is that the expressive power of GNNs scales with the number of parameters. Thus, it is hard to learn a powerful GNN when computing resources are limited. Can we build a model that enjoys the best of both worlds, i.e., a model that extracts powerful features as GNNs and is easy to train and analyze like GKs?

In this paper, we give an affirmative answer to this question. Inspired by recent connections between kernel methods and over-parameterized neural networks [Arora et al., 2019b,a, Du et al., 2019, 2018, Jacot et al., 2018, Yang, 2019], we propose a class of new graph kernels, Graph Neural Tangent Kernels (GNTKs). GNTKs are equivalent to *infinitely wide* GNNs trained by gradient descent, where the word "tangent" corresponds to the training algorithm — gradient descent. While GNTKs are induced by infinitely wide GNNs, the prediction of GNTKs depends only on pairwise kernel values between graphs, for which we give an analytic formula to calculate efficiently. Therefore, GNTKs enjoy the full expressive power of GNNs, while inheriting benefits of GKs.

**Our Contributions.** First, inspired by recent connections between over-parameterized neural networks and kernel methods Jacot et al. [2018], Arora et al. [2019a], Yang [2019], we present a general recipe which translates a GNN architecture to its corresponding GNTK. This recipe works for a wide range of GNNs, including graph isomorphism network (GIN) [Xu et al., 2019a], graph convolutional network (GCN) [Kipf and Welling, 2016], and GNN with jumping knowledge [Xu et al., 2018]. Second, we conduct a theoretical analysis of GNTKs. Using the technique developed in Arora et al. [2019b], we show for a broad range of smooth functions over graphs, a certain GNTK can learn them with polynomial number of samples. To our knowledge, this is the first sample complexity analysis in the GK and GNN literature. Finally, we validate the performance of GNTKs on 7 standard benchmark graph classification datasets. On four of them, we find GNTK outperforms all baseline methods and achieves state-of-the-art performance. In particular, GNKs achieve 83.6% accuracy on COLLAB dataset and 67.9% accuracy on PTC dataset, compared to the best of baselines, 81.0% and 64.6% respectively. Moreover, in our experiments, we also observe that GNTK is more computationally efficient than its GNN counterpart.

This paper is organized as follow. In Section 2, we provide necessary background and review operations in GNNs that we will use to derive GNTKs. In Section 3, we present our general recipe that translates a GNN to its corresponding GNTK. In Section 4, we give our theoretical analysis of GNTKs. In Section 5, we compare GNTK with state-of-the-art methods on graph classification datasets. We defer technical proofs to the supplementary material.

## 2 Preliminaries

We begin by summarizing the most common models for learning with graphs and, along the way, introducing our notation. Let $G = (V, E)$ be a graph with node features $\boldsymbol{h}_v \in \mathbb{R}^d$ for each $v \in V$. We denote the neighborhood of node $v$ by $\mathcal{N}(v)$. In this paper, we consider the graph classification task, where, given a set of graphs $\{G_1, ..., G_n\} \subseteq \mathcal{G}$ and their labels $\{y_1, ..., y_n\} \subseteq \mathcal{Y}$, our goal is to learn to predict labels of unseen graphs.

**Graph Neural Network.** GNN is a powerful framework for graph representation learning. Modern GNNs generally follow a neighborhood aggregation scheme Xu et al. [2019a], Gilmer et al. [2017],

Xu et al. [2018], where the representation $\boldsymbol{h}_v^{(\ell)}$ of each node $v$ (in layer $\ell$) is recursively updated by aggregating and transforming the representations of its neighbors. After iterations of aggregation, the representation of an entire graph is then obtained through pooling, e.g., by summing the representations of all nodes in the graph. Many GNNs, with different aggregation and graph readout functions, have been proposed under the neighborhood aggregation framework Xu et al. [2019a,b, 2018], Scarselli et al. [2009], Li et al. [2016], Kearnes et al. [2016], Ying et al. [2018], Velickovic et al. [2018], Hamilton et al. [2017], Duvenaud et al. [2015], Kipf and Welling [2016], Defferrard et al. [2016], Santoro et al. [2018, 2017], Battaglia et al. [2016].

Next, we formalize the GNN framework. We refer to the neighbor aggregation process as a BLOCK operation, and to graph-level pooling to as a READOUT operation.

**BLOCK Operation.** A BLOCK operation aggregates features over a neighborhood $\mathcal{N}(u) \cup \{u\}$ via, e.g., summation, and transforms the aggregated features with non-linearity, e.g. multi-layer perceptron (MLP) or a fully-connected layer followed by ReLU. We denote the number of fully-connected layers in each BLOCK operation, i.e., the number of hidden layers of an MLP, by $R$.

When $R = 1$, the BLOCK operation can be formulated as

$$\text{BLOCK}^{(\ell)}(u) = \sqrt{\frac{c_\sigma}{m}} \cdot \sigma \left( \boldsymbol{W}_\ell \cdot c_u \sum_{v \in \mathcal{N}(u) \cup \{u\}} \boldsymbol{h}_v^{(\ell-1)} \right).$$

Here, $\boldsymbol{W}_\ell$ are learnable weights, initialized as Gaussian random variables. $\sigma$ is an activation function like ReLU. $m$ is the output dimension of $\boldsymbol{W}_\ell$. We set the scaling factor $c_\sigma$ to 2, following the initialization scheme in He et al. [2015]. $c_u$ is a scaling factor for neighbor aggregation. Different GNNs often have different choices for $c_u$. In Graph Convolution Network (GCN) [Kipf and Welling, 2016], $c_u = \frac{1}{|\mathcal{N}(u)|+1}$, and in Graph Isomorphism Network (GIN) [Xu et al., 2019a], $c_u = 1$, which correspond to averaging and summing over neighbor features, respectively.

When the number of fully-connected layers $R = 2$, the BLOCK operation can be written as

$$\text{BLOCK}^{(\ell)}(u) = \sqrt{\frac{c_\sigma}{m}} \sigma \left( \boldsymbol{W}_{\ell,2} \sqrt{\frac{c_\sigma}{m}} \cdot \sigma \left( \boldsymbol{W}_{\ell,1} \cdot c_u \sum_{v \in \mathcal{N}(u) \cup \{u\}} \boldsymbol{h}_v^{(\ell-1)} \right) \right),$$

where $\boldsymbol{W}_{\ell,1}$ and $\boldsymbol{W}_{\ell,2}$ are learnable weights. Notice that here we first aggregate features over neighborhood $\mathcal{N}(u) \cup \{u\}$ and then transforms the aggregated features with an MLP with $R = 2$ hidden layers. BLOCK operations can be defined similarly for $R > 2$. Notice that the BLOCK operation we defined above is also known as the graph (spatial) convolutional layer in the GNN literature.

**READOUT Operation.** To get the representation of an entire graph $\boldsymbol{h}_G$ after $L$ steps of aggregation, we take the summation over all node features, i.e.,

$$\boldsymbol{h}_G = \text{READOUT} \left( \left\{ \boldsymbol{h}_u^{(L)}, u \in V \right\} \right) = \sum_{u \in V} \boldsymbol{h}_u^{(L)}.$$

There are more sophisticated READOUT operations than a simple summation Xu et al. [2018], Zhang et al. [2018a], Ying et al. [2018]. Jumping Knowledge Network (JK-Net) Xu et al. [2018] considers graph structures of different granularity, and aggregates graph features across all layers as

$$\boldsymbol{h}_G = \text{READOUT}^{\text{JK}} \left( \left\{ \boldsymbol{h}_u^{(\ell)}, u \in V, \ell \in [L] \right\} \right) = \sum_{u \in V} \left[ \boldsymbol{h}_u^{(0)}; \ldots; \boldsymbol{h}_u^{(L)} \right].$$

**Building GNNs using BLOCK and READOUT.** Most modern GNNs are constructed using the BLOCK operation and the READOUT operation Xu et al. [2019a]. We denote the number of BLOCK operations (aggregation steps) in a GNN by $L$. For each $\ell \in [L]$ and $u \in V$, we define $\boldsymbol{h}_u^{(\ell)} = \text{BLOCK}^{(\ell)}(u)$. The graph-level feature is then $\boldsymbol{h}_G = \text{READOUT} \left( \left\{ \boldsymbol{h}_u^{(L)}, u \in V \right\} \right)$ or $\boldsymbol{h}_G = \text{READOUT}^{\text{JK}} \left( \left\{ \boldsymbol{h}_u^{(\ell)}, u \in V, \ell \in [L] \right\} \right)$, depending on whether jumping knowledge (JK) is applied or not.

# 3 GNTK Formulas

In this section we present our general recipe which translates a GNN architecture to its corresponding GNTK. We first provide some intuitions on neural tangent kernels (NTKs). We refer readers to Jacot et al. [2018], Arora et al. [2019a] for more comprehensive descriptions.

## 3.1 Intuition of the Formulas

Consider a general neural network $f(\theta, x) \in \mathbb{R}$ where $\theta \in \mathbb{R}^m$ is all the parameters in the network and $x$ is the input. Given a training dataset $\{(x_i, y_i)_{i=1}^n\}$, consider training the neural network by minimizing the squared loss over training data

$$\ell(\theta) = \frac{1}{2} \sum_{i=1}^n (f(\theta, x_i) - y_i)^2.$$

Suppose we minimize the squared loss $\ell(\theta)$ by gradient descent with infinitesimally small learning rate, i.e., $\frac{d\theta(t)}{dt} = -\nabla \ell(\theta(t))$. Let $u(t) = (f(\theta(t), x_i))_{i=1}^n$ be the network outputs. $u(t)$ follows the evolution

$$\frac{du}{dt} = -\boldsymbol{H}(t)(u(t) - y),$$

where

$$\boldsymbol{H}(t)_{ij} = \left\langle \frac{\partial f(\theta(t), x_i)}{\partial \theta}, \frac{\partial f(\theta(t), x_j)}{\partial \theta} \right\rangle \text{ for } (i, j) \in [n] \times [n].$$

Recent advances in optimization of neural networks have shown, for sufficiently over-parameterized neural networks, the matrix $\boldsymbol{H}(t)$ keeps almost unchanged during the training process Arora et al. [2019b,a], Du et al. [2019, 2018], Jacot et al. [2018], in which case the training dynamics is identical to that of kernel regression. Moreover, under a random initialization of parameters, the random matrix $\boldsymbol{H}(0)$ converges in probability to a certain deterministic kernel matrix, which is called Neural Tangent Kernel (NTK) Jacot et al. [2018] and corresponds to infinitely wide neural networks. See Figure 4 in the supplementary material for an illustration.

Explicit formulas for NTKs of fully-connected neural networks have been given in Jacot et al. [2018]. Recently, explicit formulas for NTKs of convolutional neural networks are given in Arora et al. [2019a]. The goal of this section is to give an explicit formula for NTKs that correspond to GNNs defined in Section 2. Our general strategy is inspired by Arora et al. [2019a]. Let $f(\theta, G) \in \mathbb{R}$ be the output of the corresponding GNN under parameters $\theta$ and input graph $G$, for two given graphs $G$ and $G'$, to calculate the corresponding GNTK value, we need to calculate the expected value of

$$\left\langle \frac{\partial f(\theta, G)}{\partial \theta}, \frac{\partial f(\theta, G')}{\partial \theta} \right\rangle$$

in the limit that $m \to \infty$ and $\theta$ are all Gaussian random variables, which can be viewed as a Gaussian process. For each layer in the GNN, we use $\boldsymbol{\Sigma}$ to denote the covariance matrix of outputs of that layer, and $\dot{\boldsymbol{\Sigma}}$ to denote the covariance matrix corresponds to the derivative of that layer. Due to the multi-layer structure of GNNs, these covariance matrices can be naturally calculated via dynamic programming.

## 3.2 Formulas for Calculating GNTKs

Given two graphs $G = (V, E), G' = (V', E')$ with $|V| = n, |V'| = n'$ and a GNN with $L$ BLOCK operations and $R$ fully-connected layers with ReLU activation in each BLOCK operation. We give the GNTK formula of pairwise kernel value $\Theta(G, G') \in \mathbb{R}$ induced by this GNN.

We first define the covariance matrix between input features of two input graphs $G, G'$, which we use $\boldsymbol{\Sigma}^{(0)}(G, G') \in \mathbb{R}^{n \times n'}$ to denote. For two nodes $u \in V$ and $u' \in V'$, $\left[\boldsymbol{\Sigma}^{(0)}(G, G')\right]_{uu'}$ is defined to be $\boldsymbol{h}_u^\top \boldsymbol{h}_{u'}$, where $\boldsymbol{h}_u$ and $\boldsymbol{h}_{u'}$ are the input features of $u \in V$ and $u' \in V'$.

**BLOCK Operation.** A BLOCK operation in GNTK calculates a covariance matrix $\boldsymbol{\Sigma}_{(R)}^{(\ell)}(G, G') \in \mathbb{R}^{n \times n'}$ using $\boldsymbol{\Sigma}_{(R)}^{(\ell-1)}(G, G') \in \mathbb{R}^{n \times n'}$, and calculates intermediate kernel values $\boldsymbol{\Theta}_{(r)}^{(\ell)}(G, G') \in \mathbb{R}^{n \times n'}$, which will be later used to compute the final output.

More specifically, we first perform a neighborhood aggregation operation

$$\left[\boldsymbol{\Sigma}_{(0)}^{(\ell)}(G, G')\right]_{uu'} = c_u c_{u'} \sum_{v \in \mathcal{N}(u) \cup \{u\}} \sum_{v' \in \mathcal{N}(u') \cup \{u'\}} \left[\boldsymbol{\Sigma}_{(R)}^{(\ell-1)}(G, G')\right]_{vv'},$$

$$\left[\boldsymbol{\Theta}_{(0)}^{(\ell)}(G, G')\right]_{uu'} = c_u c_{u'} \sum_{v \in \mathcal{N}(u) \cup \{u\}} \sum_{v' \in \mathcal{N}(u') \cup \{u'\}} \left[\boldsymbol{\Theta}_{(R)}^{(\ell-1)}(G, G')\right]_{vv'}.$$

Here we define $\boldsymbol{\Sigma}_{(R)}^{(0)}(G, G')$ and $\boldsymbol{\Theta}_{(R)}^{(0)}(G, G')$ as $\boldsymbol{\Sigma}^{(0)}(G, G')$, for notational convenience. Next we perform $R$ transformations that correspond to the $R$ fully-connected layers with ReLU activation. Here $\sigma(z) = \max\{0, z\}$ is the ReLU activation function. We denote $\dot{\sigma}(z) = \mathbb{1}[z \geq 0]$ to be the derivative of the ReLU activation function.

For each $r \in [R]$, we define

- For $u \in V, u' \in V'$,

$$\left[\boldsymbol{A}_{(r)}^{(\ell)}(G, G')\right]_{uu'} = \begin{pmatrix} \left[\boldsymbol{\Sigma}_{(r-1)}^{(\ell)}(G, G)\right]_{u,u} & \left[\boldsymbol{\Sigma}_{(r-1)}^{(\ell)}(G, G')\right]_{uu'} \\ \left[\boldsymbol{\Sigma}_{(r-1)}^{(\ell)}(G', G)\right]_{uu'} & \left[\boldsymbol{\Sigma}_{(r-1)}^{(\ell)}(G', G')\right]_{u'u'} \end{pmatrix} \in \mathbb{R}^{2 \times 2}.$$

- For $u \in V, u' \in V'$,

$$\left[\boldsymbol{\Sigma}_{(r)}^{(\ell)}(G, G')\right]_{uu'} = c_\sigma \mathbb{E}_{(a,b) \sim \mathcal{N}\left(\boldsymbol{0}, \left[\boldsymbol{A}_{(r)}^{(\ell)}(G,G')\right]_{uu'}\right)} \left[\sigma(a)\, \sigma(b)\right], \tag{1}$$

$$\left[\dot{\boldsymbol{\Sigma}}_{(r)}^{(\ell)}(G, G')\right]_{uu'} = c_\sigma \mathbb{E}_{(a,b) \sim \mathcal{N}\left(\boldsymbol{0}, \left[\boldsymbol{A}_{(r)}^{(\ell)}(G,G')\right]_{uu'}\right)} \left[\dot{\sigma}(a)\dot{\sigma}(b)\right]. \tag{2}$$

- For $u \in V, u' \in V'$,

$$\left[\boldsymbol{\Theta}_{(r)}^{(\ell)}(G, G')\right]_{uu'} = \left[\boldsymbol{\Theta}_{(r-1)}^{(\ell)}(G, G')\right]_{uu'} \left[\dot{\boldsymbol{\Sigma}}_{(r)}^{(\ell)}(G, G')\right]_{uu'} + \left[\boldsymbol{\Sigma}_{(r)}^{(\ell)}(G, G')\right]_{uu'}.$$

Note in the above we have shown how to calculate $\boldsymbol{\Theta}_{(R)}^{(\ell)}(G, G')$ for each $\ell \in \{0, 1, \ldots, L\}$. These intermediate outputs will be used to calculate the final output of the corresponding GNTK.

**READOUT Operation.** Given these intermediate outputs, we can now calculate the final output of GNTK using the following formula.

$$\Theta(G, G') = \begin{cases} \sum_{u \in V, u' \in V'} \left[\boldsymbol{\Theta}_{(R)}^{(\ell)}(G, G')\right]_{uu'} & \text{without jumping knowledge} \\ \sum_{u \in V, u' \in V'} \left[\sum_{\ell=0}^{L} \boldsymbol{\Theta}_{(R)}^{(\ell)}(G, G')\right]_{uu'} & \text{with jumping knowledge} \end{cases}.$$

To better illustrate our general recipe, in Figure 1 we give a concrete example in which we translate a GNN with $L = 2$ BLOCK operations, $R = 1$ fully-connection layer in each BLOCK operation, and jumping knowledge, to its corresponding GNTK.

## 4  Theoretical Analysis of GNTK

In this section, we analyze the generalization ability of a GNTK that corresponds to a simple GNN. We consider the standard supervised learning setup. We are given $n$ training data $\{(G_i, y_i)\}_{i=1}^{n}$ drawn i.i.d. from the underlying distribution $\mathcal{D}$, where $G_i$ is the $i$-th input graph and $y_i$ is its label. Consider a GNN with a single BLOCK operation, followed by the READOUT operation (without jumping knowledge). Here we set $c_u = \left(\left\|\sum_{v \in \mathcal{N}(u) \cup \{u\}} \boldsymbol{h}_v\right\|_2\right)^{-1}$. We use $\boldsymbol{\Theta} \in \mathbb{R}^{n \times n}$ to denote the kernel matrix, where $\boldsymbol{\Theta}_{ij} = \Theta(G_i, G_j)$. Here $\Theta(G, G')$ is the kernel function that corresponds

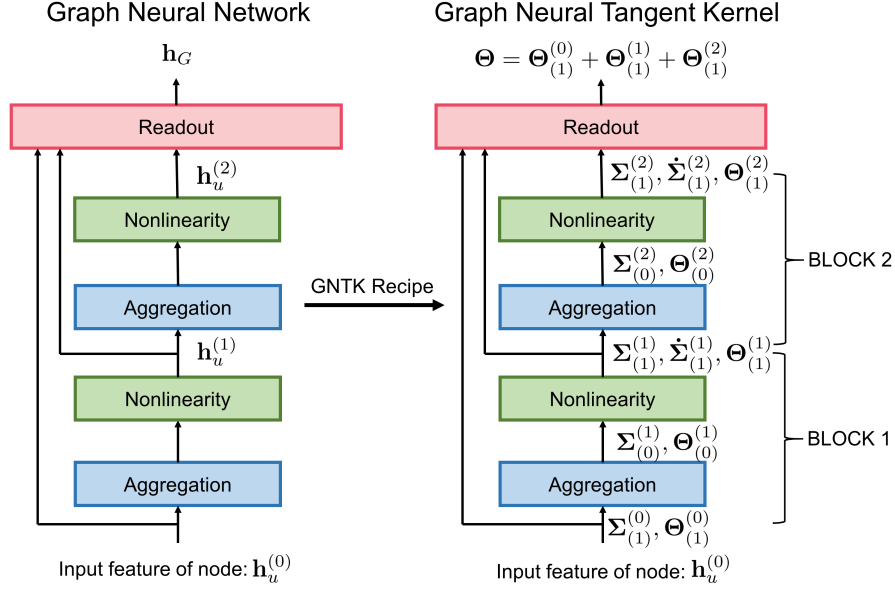

Figure 1: **Illustration of our recipe that translates a GNN to a GNTK.** For a GNN with $L = 2$ BLOCK operations, $R = 1$ fully-connected layer in each BLOCK operation, and jumping knowledge, the corresponding GNTK is calculated as follow. For two graphs $G$ and $G'$, we first calculate $\left[\boldsymbol{\Theta}^{(0)}_{(1)}(G, G')\right]_{uu'} = \left[\boldsymbol{\Sigma}^{(0)}_{(1)}(G, G')\right]_{uu'} = \left[\boldsymbol{\Sigma}^{(0)}(G, G')\right]_{uu'} = \boldsymbol{h}_u^\top \boldsymbol{h}_{u'}$. We follow the kernel formulas in Section 3 to calculate $\boldsymbol{\Sigma}^{(\ell)}_{(0)}, \boldsymbol{\Theta}^{(\ell)}_{(0)}$ using $\boldsymbol{\Sigma}^{(\ell-1)}_{(R)}, \boldsymbol{\Theta}^{(\ell-1)}_{(R)}$ (Aggregation) and calculate $\boldsymbol{\Sigma}^{(\ell)}_{(r)}, \dot{\boldsymbol{\Sigma}}^{(\ell)}_{(r)}, \boldsymbol{\Theta}^{(\ell)}_{(r)}$ using $\boldsymbol{\Sigma}^{(\ell)}_{(r-1)}, \boldsymbol{\Theta}^{(\ell)}_{(r-1)}$ (Nonlinearity). The final output is $\Theta(G, G') = \sum_{u \in V, u' \in V'} \left[\sum_{\ell=0}^{L} \boldsymbol{\Theta}^{(\ell)}_{(R)}(G, G')\right]_{uu'}$.

to the simple GNN. See Section 3 for the formulas for calculating $\Theta(G, G')$. Throughout the discussion, we assume that the kernel matrix $\boldsymbol{\Theta} \in \mathbb{R}^{n \times n}$ is invertible.

For a testing point $G_{te}$, the prediction of kernel regression using GNTK on this testing point is

$$f_{ker}(G_{te}) = [\Theta(G_{te}, G_1), \Theta(G_{te}, G_1), \dots, \Theta(G_{te}, G_n)]^\top \boldsymbol{\Theta}^{-1} \boldsymbol{y}.$$

The following result is a standard result for kernel regression proved using Rademacher complexity. For a proof, see Bartlett and Mendelson [2002].

**Theorem 4.1** (Bartlett and Mendelson [2002])**.** *Given $n$ training data $\{(G_i, y_i)\}_{i=1}^n$ drawn i.i.d. from the underlying distribution $\mathcal{D}$. Consider any loss function $\ell : \mathbb{R} \times \mathbb{R} \to [0, 1]$ that is 1-Lipschitz in the first argument such that $\ell(y, y) = 0$. With probability at least $1 - \delta$, the population loss of the GNTK predictor can be upper bounded by*

$$L_{\mathcal{D}}(f_{ker}) = \mathbb{E}_{(G,y) \sim \mathcal{D}}[\ell(f_{ker}(G), y)] = O\left(\frac{\sqrt{\boldsymbol{y}^\top \boldsymbol{\Theta}^{-1} \boldsymbol{y} \cdot \text{tr}(\boldsymbol{\Theta})}}{n} + \sqrt{\frac{\log(1/\delta)}{n}}\right).$$

Note that this theorem presents a data-dependent generalization bound which is related to the kernel matrix $\boldsymbol{\Theta} \in \mathbb{R}^{n \times n}$ and the labels $\{y_i\}_{i=1}^n$. Using this theorem, if we can bound $\boldsymbol{y}^\top \boldsymbol{\Theta}^{-1} \boldsymbol{y}$ and $\text{tr}(\boldsymbol{\Theta})$, then we can obtain a concrete sample complexity bound. We instantiate this idea to study the class of graph labeling functions that can be efficiently learned by GNTKs.

The following two theorems guarantee that if labels are generated as described in (3), then the GNTK that corresponds to the simple GNN described above can learn this function with polynomial number of samples. We first give an upper bound on $\boldsymbol{y}^\top \boldsymbol{\Theta}^{-1} \boldsymbol{y}$.

**Theorem 4.2.** *For each $i \in [n]$, if the labels $\{y_i\}_{i=1}^n$ satisfy*

$$y_i = \alpha_1 \sum_{u \in V} \left( \overline{\boldsymbol{h}}_u^\top \boldsymbol{\beta}_1 \right) + \sum_{l=1}^{\infty} \alpha_{2l} \sum_{u \in V} \left( \overline{\boldsymbol{h}}_u^\top \boldsymbol{\beta}_{2l} \right)^{2l}, \qquad (3)$$

*where $\overline{\boldsymbol{h}}_u = c_u \sum_{v \in \mathcal{N}(u) \cup \{u\}} \boldsymbol{h}_v$, $\alpha_1, \alpha_2, \alpha_4, \ldots \in \mathbb{R}$, $\boldsymbol{\beta}_1, \boldsymbol{\beta}_2, \boldsymbol{\beta}_4, \ldots \in \mathbb{R}^d$, and $G_i = (V, E)$, then we have*

$$\sqrt{\boldsymbol{y}^\top \boldsymbol{\Theta}^{-1} \boldsymbol{y}} \le 2|\alpha_1| \cdot \|\boldsymbol{\beta}_1\|_2 + \sum_{l=1}^{\infty} \sqrt{2\pi}(2l-1)|\alpha_{2l}| \cdot \|\boldsymbol{\beta}_{2l}\|_2^{2l}.$$

The following theorem gives an upper bound on $\text{tr}(\boldsymbol{\Theta})$.

**Theorem 4.3.** *If for all graphs $G_i = (V_i, E_i)$ in the training set, $|V_i|$ is upper bounded by $\overline{V}$, then $\text{tr}(\boldsymbol{\Theta}) \le O(n\overline{V}^2)$. Here $n$ is the number of training samples.*

Combining Theorem 4.2 and Theorem 4.3 with Theorem 4.1, we know if

$$2|\alpha_1| \cdot \|\boldsymbol{\beta}_1\|_2 + \sum_{l=1}^{\infty} \sqrt{2\pi}(2l-1)|\alpha_{2l}| \cdot \|\boldsymbol{\beta}_{2l}\|_2^{2l}$$

is bounded, and $|V_i|$ is bounded for all graphs $G_i = (V_i, E_i)$ in the training set, then the GNTK that corresponds to the simple GNN described above can learn functions of forms in (3), with polynomial number of samples. To our knowledge, this is the first sample complexity analysis in the GK and GNN literature.

## 5   Experiments

In this section, we demonstrate the effectiveness of GNTKs using experiments on graph classification tasks. For ablation study, we investigate how the performance varies with the architecture of the corresponding GNN. Following common practices of evaluating performance of graph classification models Yanardag and Vishwanathan [2015], we perform 10-fold cross validation and report the mean and standard deviation of validation accuracies. More details about the experiment setup can be found in Section B of the supplementary material.

**Datasets.**   The benchmark datasets include four bioinformatics datasets MUTAG, PTC, NCI1, PROTEINS and three social network datasets COLLAB, IMDB-BINARY, IMDB-MULTI. For each graph, we transform the categorical input features to one-hot encoding representations. For datasets where the graphs have no node features, i.e. only graph structure matters, we use degrees as input node features.

### 5.1   Results

We compare GNTK with various state-of-the-art graph classification algorithms: (1) the WL subtree kernel Shervashidze et al. [2011]; (2) state-of-the-art deep learning architectures, including Graph Convolutional Network (GCN) Kipf and Welling [2016], GraphSAGE Hamilton et al. [2017], Graph Isomorphism Network(GIN) Xu et al. [2019a], PATCHY-SANNiepert et al. [2016] and Deep Graph CNN (DGCNN) Zhang et al. [2018a]; (3) Graph kernels based on random walks, i.e., Anonymous Walk Embeddings Ivanov and Burnaev [2018] and RetGK Zhang et al. [2018b]. For deep learning methods and random walk graph kernels, we report the accuracies reported in the original papers. The experiment setup is deferred to Section B.

The graph classification results are shown in Table 1. The best results are highlighted as bold. Our proposed GNTKs are powerful and achieve state-of-the-art classification accuracy on most datasets. In four of them, we find GNTKs outperform all baseline methods. In particular, GNTKs achieve 83.6% accuracy on COLLAB dataset and 67.9% accuracy on PTC dataset, compared to the best of baselines, 81.0% and 64.6% respectively. Notably, GNTKs give the best performance on all social network datasets. Moreover, In our experiments, we also observe that with the same architecture, GNTK is more computational efficient that its GNN counterpart. On IMDB-B dataset, running GIN with the default setup (official implementation of Xu et al. [2019a]) takes 19 minutes on a TITAN X GPU and running GNTK only takes 2 minutes.

|  | Method | COLLAB | IMDB-B | IMDB-M | PTC | NCI1 | MUTAG | PROTEINS |
|---|---|---|---|---|---|---|---|---|
| GNN | GCN | $79.0 \pm 1.8$ | $74.0 \pm 3.4$ | $51.9 \pm 3.8$ | $64.2 \pm 4.3$ | $80.2 \pm 2.0$ | $85.6 \pm 5.8$ | $76.0 \pm 3.2$ |
|  | GraphSAGE | – | $72.3 \pm 5.3$ | $50.9 \pm 2.2$ | $63.9 \pm 7.7$ | $77.7 \pm 1.5$ | $85.1 \pm 7.6$ | $75.9 \pm 3.2$ |
|  | PatchySAN | $72.6 \pm 2.2$ | $71.0 \pm 2.2$ | $45.2 \pm 2.8$ | $60.0 \pm 4.8$ | $78.6 \pm 1.9$ | $\mathbf{92.6 \pm 4.2}$ | $75.9 \pm 2.8$ |
|  | DGCNN | $73.7$ | $70.0$ | $47.8$ | $58.6$ | $74.4$ | $85.8$ | $75.5$ |
|  | GIN | $80.2 \pm 1.9$ | $75.1 \pm 5.1$ | $52.3 \pm 2.8$ | $64.6 \pm 7.0$ | $82.7 \pm 1.7$ | $89.4 \pm 5.6$ | $\mathbf{76.2 \pm 2.8}$ |
| GK | WL subtree | $78.9 \pm 1.9$ | $73.8 \pm 3.9$ | $50.9 \pm 3.8$ | $59.9 \pm 4.3$ | $\mathbf{86.0 \pm 1.8}$ | $90.4 \pm 5.7$ | $75.0 \pm 3.1$ |
|  | AWL | $73.9 \pm 1.9$ | $74.5 \pm 5.9$ | $51.5 \pm 3.6$ | – | – | $87.9 \pm 9.8$ | – |
|  | RetGK | $81.0 \pm 0.3$ | $71.9 \pm 1.0$ | $47.7 \pm 0.3$ | $62.5 \pm 1.6$ | $84.5 \pm 0.2$ | $90.3 \pm 1.1$ | $75.8 \pm 0.6$ |
|  | GNTK | $\mathbf{83.6 \pm 1.0}$ | $\mathbf{76.9 \pm 3.6}$ | $\mathbf{52.8 \pm 4.6}$ | $\mathbf{67.9 \pm 6.9}$ | $84.2 \pm 1.5$ | $90.0 \pm 8.5$ | $75.6 \pm 4.2$ |

Table 1: **Classification results (in %) for graph classification datasets.** GNN: graph neural network based methods. GK: graph kernel based methods. GNTK: fusion of GNN and GK.

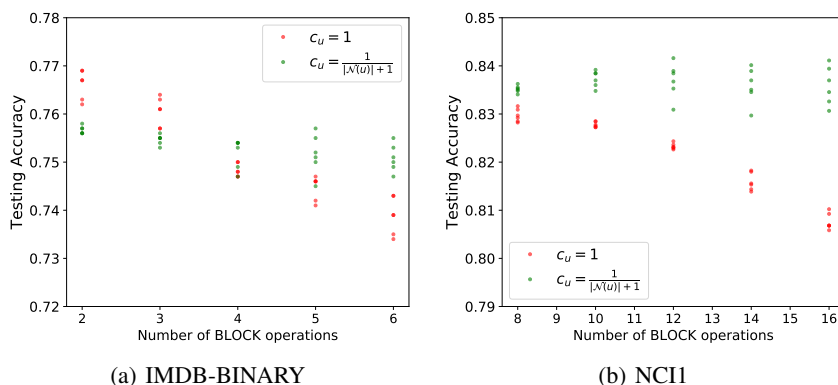

(a) IMDB-BINARY  (b) NCI1

Figure 2: **Effects of number of BLOCK operations and the scaling factor $c_u$ on the performance of GNTK.** Each dot represents the performance of a particular GNTK architecture. We divide different architectures into different groups by number of BLOCK operations. We color these GNTK architecture points by the scaling factor $c_u$. We observe the test accuracy is correlated with the dataset and the architecture.

## 5.2 Relation between GNTK Performance and the Corresponding GNN

We conduct ablation study to investigate how the performance of GNTK varies as we change the architecture of the corresponding GNN. We select two representative datasets, one social network dataset IMDBBINARY, and another bioinformatics dataset NCI1. For IMDBBINARY, we vary the number of BLOCK operations in $\{2, 3, 4, 5, 6\}$. For NCI1, we vary the number of BLOCK operations in $\{8, 10, 12, 14, 16\}$. For both datasets, we vary the number of MLP layers in $\{1, 2, 3\}$.

**Effects of Number of BLOCK Operations and the Scaling Factor $c_u$.** We investigate how the performance of GNTKs is correlated with number of BLOCK operations and the scaling factor $c_u$. First, on the bioinformatics dataset (NCI), we observe that GNTKs with more layers perform better. This is perhaps because, for molecules and bio graphs, more global structural information is helpful, as they provide important information about the chemical/bio entity. On such graphs, GNTKs are particularly effective because GNTKs can easily scale to many layers, whereas the number of layers in GNNs may be restricted by computing resources.

Moreover, the performance of GNTK is correlated with that of the corresponding GNN. For example, in social networks, GNTKs with sum aggregation $c_u = 1$ work better than average aggregation $c_u = \frac{1}{|\mathcal{N}(u)|+1}$. The similar pattern holds in GNNs, because sum aggregation learns more graph structure information than average aggregation Xu et al. [2019a]. This suggests GNTK can indeed inherit the properties and advantages of the corresponding GNN, while also gaining the benefits of graph kernels.

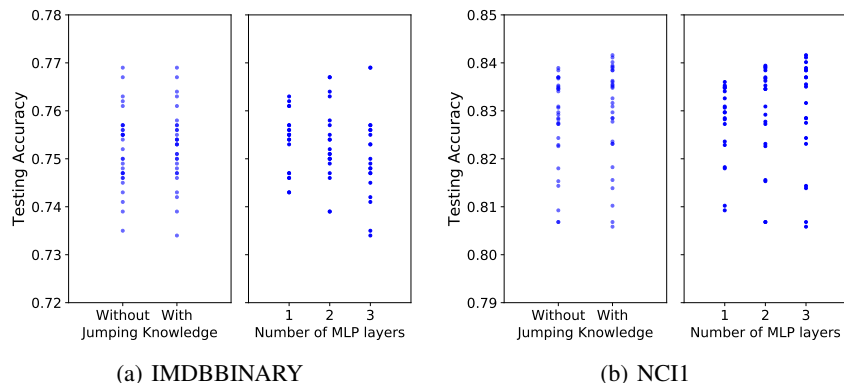

(a) IMDBBINARY               (b) NCI1

Figure 3: **Effects of jumping knowledge and number of MLP layers on the performance of GNTK.** Each dot represents the test performance of a GNTK architecture. We divide all GNTK architectures into different groups, according to whether jumping knowledge is applied, or number of MLP layers.

**Effects of Jumping Knowledge and Number of MLP Layers**   In the GNN literature, jumping knowledge network (JK) is expected to improve performance Xu et al. [2018], Fey [2019]. In Figure 3, we observe that a similar trend holds for GNTK. The performance of GNTK is improved on both NCI and IMDB datasets when jumping knowledge is applied. Moreover, increasing the number of MLP layers can increase the performance by $\sim 0.8\%$. These empirical findings further confirm that GNTKs can inherit the benefits of GNNs, since improvements on GNN architectures are reflected in the improvements GNTKs.

We conclude that GNTKs are attractive for graph representation learning because they can combine the advantages of both GNNs and GKs.

# Acknowledgments

S. S. Du and B. Póczos acknowledge support from AFRL grant FA8750-17-2-0212 and DARPA D17AP0000. R. Salakhutdinov and R. Wang are supported in part by NSF IIS-1763562, Office of Naval Research grant N000141812861, and Nvidia NVAIL award. K. Xu is supported by NSF CAREER award 1553284 and a Chevron-MIT Energy Fellowship. This work was performed while S. S. Du was a Ph.D. student at Carnegie Mellon University and K. Hou was visiting Carnegie Mellon University.

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
