[Supplementary Material]

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

Figure 4: Illustration of NTK theory. Consider a general neural network with $L$ layers $\theta^{(1)}, \ldots, \theta^{(L)}$, given input $x_i$ and $x_j$, the neural network will output $f(\theta(t), x_i), f(\theta(t), x_j)$. When trained by gradient descent, evolution of $u(t)$ follows $\frac{du}{dt} = -\boldsymbol{H}(t)(u(t) - y)$. One can show that when the number of parameters in the neural network is large enough, and parameters of the neural network are initialized as Gaussian variables, $\boldsymbol{H}(t) \approx \boldsymbol{H}(0)$ and can be calculated analytically.

## A    Missing Proofs

### A.1    Proof of Theorem 4.2

*Proof.* By Section 3, for two graph $G$ and $G'$, the GNTK kernel function that corresponds to the simple GNN can be described as

$$\Theta(G, G') = \sum_{u \in V, u' \in V'} \left( \left[\boldsymbol{\Sigma}_{(0)}^{(1)}(G, G')\right]_{uu'} \left[\dot{\boldsymbol{\Sigma}}_{(1)}^{(1)}(G, G')\right]_{uu'} + \left[\boldsymbol{\Sigma}_{(1)}^{(1)}(G, G')\right]_{uu'} \right).$$

Here, we have

$$\left[\boldsymbol{\Sigma}_{(0)}^{(1)}(G, G')\right]_{uu'} = c_u c_{u'} \left( \sum_{v \in \mathcal{N}(u) \cup \{u\}} \boldsymbol{h}_v \right)^{\top} \left( \sum_{v' \in \mathcal{N}(u') \cup \{u'\}} \boldsymbol{h}_{v'} \right) = \overline{\boldsymbol{h}}_u^{\top} \overline{\boldsymbol{h}}_{u'}.$$

Recall that

$$\left[\boldsymbol{\Sigma}_{(r)}^{(\ell)}(G, G')\right]_{uu'} = c_\sigma \mathbb{E}_{(a,b) \sim \mathcal{N}\left(\boldsymbol{0}, \left[\boldsymbol{A}_{(r)}^{(\ell)}(G, G')\right]_{uu'}\right)} \left[\sigma(a)\sigma(b)\right],$$

$$\left[\dot{\boldsymbol{\Sigma}}_{(r)}^{(\ell)}(G, G')\right]_{uu'} = c_\sigma \mathbb{E}_{(a,b) \sim \mathcal{N}\left(\boldsymbol{0}, \left[\boldsymbol{A}_{(r)}^{(\ell)}(G, G')\right]_{uu'}\right)} \left[\dot{\sigma}(a)\dot{\sigma}(b)\right]$$

and

$$\left[\boldsymbol{A}_{(r)}^{(\ell)}(G, G')\right]_{uu'} = \begin{pmatrix} \left[\boldsymbol{\Sigma}_{(r-1)}^{(\ell)}(G, G)\right]_{u,u} & \left[\boldsymbol{\Sigma}_{(r-1)}^{(\ell)}(G, G')\right]_{uu'} \\ \left[\boldsymbol{\Sigma}_{(r-1)}^{(\ell)}(G', G)\right]_{uu'} & \left[\boldsymbol{\Sigma}_{(r-1)}^{(\ell)}(G', G')\right]_{u'u'} \end{pmatrix} \in \mathbb{R}^{2 \times 2}.$$

Since $\sigma(z) = \max\{0, z\}$ is the ReLU activation function, and $\dot{\sigma}(z) = \mathbb{1}[z \geq 0]$ is the derivative of the ReLU activation function, and $\|\overline{\boldsymbol{h}}_u\|_2 = 1$ for all nodes $u$, by calculation, we have

$$\left[\dot{\boldsymbol{\Sigma}}_{(1)}^{(1)}(G, G')\right]_{uu'} = \frac{\pi - \arccos\left(\left[\boldsymbol{\Sigma}_{(0)}^{(1)}(G, G')\right]_{uu'}\right)}{2\pi},$$

$$\left[\mathbf{\Sigma}^{(1)}_{(1)}(G,G')\right]_{uu'} = \frac{\pi - \arccos\left(\left[\mathbf{\Sigma}^{(1)}_{(0)}(G,G')\right]_{uu'}\right) + \sqrt{1 - \left[\mathbf{\Sigma}^{(1)}_{(0)}(G,G')\right]^2_{uu'}}}{2\pi}.$$

Since
$$\arcsin(x) = \sum_{l=0}^{\infty} \frac{(2l-1)!!}{(2l)!!} \cdot \frac{x^{2l+1}}{2l+1},$$

we have
$$\left[\mathbf{\Sigma}^{(1)}_{(0)}(G,G')\right]_{uu'}\left[\dot{\mathbf{\Sigma}}^{(1)}_{(1)}(G,G')\right]_{uu'} = \frac{1}{4}\left[\mathbf{\Sigma}^{(1)}_{(0)}(G,G')\right]_{uu'} + \frac{1}{2\pi}\left[\mathbf{\Sigma}^{(1)}_{(0)}(G,G')\right]_{uu'}\arcsin\left(\left[\mathbf{\Sigma}^{(1)}_{(0)}(G,G')\right]_{uu'}\right)$$
$$= \frac{1}{4}\left[\mathbf{\Sigma}^{(1)}_{(0)}(G,G')\right]_{uu'} + \frac{1}{2\pi}\sum_{l=1}^{\infty}\frac{(2l-3)!!}{(2l-2)!!\cdot(2l-1)}\cdot\left[\mathbf{\Sigma}^{(1)}_{(0)}(G,G')\right]^{2l}_{uu'}$$
$$= \frac{1}{4}\overline{\boldsymbol{h}}_u^{\top}\overline{\boldsymbol{h}}_{u'} + \frac{1}{2\pi}\sum_{l=1}^{\infty}\frac{(2l-3)!!}{(2l-2)!!\cdot(2l-1)}\cdot\left(\overline{\boldsymbol{h}}_u^{\top}\overline{\boldsymbol{h}}_{u'}\right)^{2l}.$$

Let $\mathbf{\Phi}^{(2l)}(\cdot)$ be the feature map of the polynomial kernel of degree $2l$, i.e.,
$$k^{(2l)}(\boldsymbol{x},\boldsymbol{y}) = \left(\boldsymbol{x}^{\top}\boldsymbol{y}\right)^{2l} = \mathbf{\Phi}^{(2l)}(\boldsymbol{x})^{\top}\mathbf{\Phi}^{(2l)}(\boldsymbol{y}).$$

We have
$$\left[\mathbf{\Sigma}^{(1)}_{(0)}(G,G')\right]_{uu'}\left[\dot{\mathbf{\Sigma}}^{(1)}_{(1)}(G,G')\right]_{uu'}$$
$$= \frac{1}{4}\overline{\boldsymbol{h}}_u^{\top}\overline{\boldsymbol{h}}_{u'} + \frac{1}{2\pi}\sum_{l=1}^{\infty}\frac{(2l-3)!!}{(2l-2)!!\cdot(2l-1)}\cdot\left(\mathbf{\Phi}^{(2l)}(\overline{\boldsymbol{h}}_u)\right)^{\top}\mathbf{\Phi}^{(2l)}(\overline{\boldsymbol{h}}_{u'}).$$

Let
$$\Theta_1(G,G') = \sum_{u\in V, u'\in V'}\left[\mathbf{\Sigma}^{(1)}_{(0)}(G,G')\right]_{uu'}\left[\dot{\mathbf{\Sigma}}^{(1)}_{(1)}(G,G')\right]_{uu'},$$

we have
$$\Theta_1(G,G') = \frac{1}{4}\left(\sum_{u\in V}\overline{\boldsymbol{h}}_u\right)^{\top}\left(\sum_{u'\in V'}\overline{\boldsymbol{h}}_{u'}\right) + \frac{1}{2\pi}\sum_{l=1}^{\infty}\frac{(2l-3)!!}{(2l-2)!!\cdot(2l-1)}\cdot\left(\sum_{u\in V}\mathbf{\Phi}^{(2l)}(\overline{\boldsymbol{h}}_u)\right)^{\top}\left(\sum_{u'\in V'}\mathbf{\Phi}^{(2l)}(\overline{\boldsymbol{h}}_{u'})\right).$$

Since $\mathbf{\Theta} = \mathbf{\Theta}_1 + \mathbf{\Theta}_2$ where $\mathbf{\Theta}_2$ is a kernel matrix (and thus positive semi-definite), for any $y \in \mathbb{R}^n$, we have
$$\boldsymbol{y}^{\top}\mathbf{\Theta}^{-1}\boldsymbol{y} \le \boldsymbol{y}^{\top}\mathbf{\Theta}_1^{-1}\boldsymbol{y}.$$

Recall that
$$y_i = \alpha_1\sum_{u\in V}\left(\overline{\boldsymbol{h}}_u^{\top}\boldsymbol{\beta}_1\right) + \sum_{l=1}^{\infty}\alpha_{2l}\sum_{u\in V}\left(\overline{\boldsymbol{h}}_u^{\top}\boldsymbol{\beta}_{2l}\right)^{2l}.$$

We rewrite
$$y_i = y_i^{(0)} + \sum_{l=1}^{\infty}y_i^{(2l)},$$

where
$$y_i^{(0)} = \alpha_1\left(\sum_{u\in V}\overline{\boldsymbol{h}}_u\right)^{\top}\boldsymbol{\beta}_1,$$

and for each $l \ge 1$,
$$y_i^{(2l)} = \alpha_{2l}\sum_{u\in V}\left(\overline{\boldsymbol{h}}_u^{\top}\boldsymbol{\beta}_{2l}\right)^{2l} = \alpha_{2l}\sum_{u\in V}\left(\mathbf{\Phi}^{2l}(\overline{\boldsymbol{h}}_u)\right)^{\top}\mathbf{\Phi}^{2l}(\boldsymbol{\beta}_{2l}) = \alpha_{2l}\left(\sum_{u\in V}\mathbf{\Phi}^{2l}(\overline{\boldsymbol{h}}_u)\right)^{\top}\mathbf{\Phi}^{2l}(\boldsymbol{\beta}_{2l}).$$

We have
$$\boldsymbol{y} = \boldsymbol{y}^{(0)} + \sum_{l=1}^{\infty} \boldsymbol{y}^{(2l)}.$$

Thus,
$$\sqrt{\boldsymbol{y}^{\top} \boldsymbol{\Theta}^{-1} \boldsymbol{y}} \leq \sqrt{\boldsymbol{y}^{\top} \boldsymbol{\Theta}_1^{-1} \boldsymbol{y}} \leq \sqrt{\left(\boldsymbol{y}^{(0)}\right)^{\top} \boldsymbol{\Theta}_1^{-1} \boldsymbol{y}^{(0)}} + \sum_{l=1}^{\infty} \sqrt{\left(\boldsymbol{y}^{(2l)}\right)^{\top} \boldsymbol{\Theta}_1^{-1} \boldsymbol{y}^{(2l)}}.$$

When $l = 0$, we have
$$\sqrt{\left(\boldsymbol{y}^{0}\right)^{\top} \boldsymbol{\Theta}_1^{-1} \boldsymbol{y}^{0}} \leq 2|\alpha_1| \|\boldsymbol{\beta}_1\|_2.$$

When $l \geq 1$, we have
$$\sqrt{\left(\boldsymbol{y}^{2l}\right)^{\top} \boldsymbol{\Theta}_1^{-1} \boldsymbol{y}^{2l}} \leq \sqrt{2\pi}(2l-1)|\alpha_{2l}| \left\|\boldsymbol{\Phi}^{2l}\left(\boldsymbol{\beta}_{2l}\right)\right\|_2.$$

Notice that
$$\left\|\boldsymbol{\Phi}^{2l}\left(\boldsymbol{\beta}_{2l}\right)\right\|_2^2 = \left(\boldsymbol{\Phi}^{2l}\left(\boldsymbol{\beta}_{2l}\right)\right)^{\top} \boldsymbol{\Phi}^{2l}\left(\boldsymbol{\beta}_{2l}\right) = \|\boldsymbol{\beta}_{2l}\|_2^{4l}.$$

Thus,
$$\sqrt{\boldsymbol{y}^{\top} \boldsymbol{\Theta}^{-1} \boldsymbol{y}} \leq 2|\alpha_1| \|\boldsymbol{\beta}_1\|_2 + \sum_{l=1}^{\infty} \sqrt{2\pi}(2l-1)|\alpha_{2l}| \|\boldsymbol{\beta}_{2l}\|_2^{2l}.$$

$\square$

## A.2   Proof of Theorem 4.3

*Proof.* Recall that
$$\Theta(G, G') = \sum_{u \in V, u' \in V'} \left( \left[\boldsymbol{\Sigma}_{(0)}^{(1)}(G, G')\right]_{uu'} \left[\dot{\boldsymbol{\Sigma}}_{(1)}^{(1)}(G, G')\right]_{uu'} + \left[\boldsymbol{\Sigma}_{(1)}^{(1)}(G, G')\right]_{uu'} \right),$$

where
$$\left[\boldsymbol{\Sigma}_{(0)}^{(1)}(G, G')\right]_{uu'} = c_u c_{u'} \left( \sum_{v \in \mathcal{N}(u) \cup \{u\}} \boldsymbol{h}_v \right)^{\top} \left( \sum_{v' \in \mathcal{N}(u') \cup \{u'\}} \boldsymbol{h}_{v'} \right) = \overline{\boldsymbol{h}}_u^{\top} \overline{\boldsymbol{h}}_{u'}$$

and
$$\left[\dot{\boldsymbol{\Sigma}}_{(1)}^{(1)}(G, G')\right]_{uu'} = \frac{\pi - \arccos\left(\left[\boldsymbol{\Sigma}_{(0)}^{(1)}(G, G')\right]_{uu'}\right)}{2\pi},$$

$$\left[\boldsymbol{\Sigma}_{(1)}^{(1)}(G, G')\right]_{uu'} = \frac{\pi - \arccos\left(\left[\boldsymbol{\Sigma}_{(0)}^{(1)}(G, G')\right]_{uu'}\right) + \sqrt{1 - \left[\boldsymbol{\Sigma}_{(0)}^{(1)}(G, G')\right]_{uu'}^2}}{2\pi}.$$

Since for each node $u$, $\overline{\boldsymbol{h}}_u = c_u \sum_{v \in \mathcal{N}(u) \cup \{u\}} \boldsymbol{h}_v$, and $c_u = \left(\left\|\sum_{v \in \mathcal{N}(u) \cup \{u\}} \boldsymbol{h}_v\right\|_2\right)^{-1}$, we have $\|\overline{\boldsymbol{h}}_u\|_2 = 1$. Moreover,
$$\left[\dot{\boldsymbol{\Sigma}}_{(1)}^{(1)}(G, G')\right]_{uu'} = \frac{\pi - \arccos\left(\left[\boldsymbol{\Sigma}_{(0)}^{(1)}(G, G')\right]_{uu'}\right)}{2\pi} \leq 1/2$$

and
$$\left[\boldsymbol{\Sigma}_{(1)}^{(1)}(G, G')\right]_{uu'} = \frac{\pi - \arccos\left(\left[\boldsymbol{\Sigma}_{(0)}^{(1)}(G, G')\right]_{uu'}\right) + \sqrt{1 - \left[\boldsymbol{\Sigma}_{(0)}^{(1)}(G, G')\right]_{uu'}^2}}{2\pi} \leq \frac{1 + \pi}{2\pi} \leq 1,$$

we have
$$\Theta(G, G') \leq 2|V||V'|.$$

Thus,
$$\mathrm{tr}(\boldsymbol{\Theta}) \leq 2n\overline{V}^2.$$

$\square$

# B  Experiment Setup

To calculate GNTKs, we adopt the formulas provided in Section 3.2. To calculate the expectation of the post-activation output, i.e., (1) and (2), we use the same approach as in Arora et al. [2019a] (cf. Section 4.3 in Arora et al. [2019a]).

For GNTKs, we tune the following hyperparameters.

1. The number of BLOCK operations. We search from candidate values $\{1, 2, \ldots, 14\}$.
2. The number of fully-connected layers in each BLOCK operation. We search from candidate values $\{1, 2, 3\}$.
3. The parameter $c_u$. We search from candidate values $\left\{1, \frac{1}{|\mathcal{N}(u)|+1}\right\}$.

To utilize the GNTKs we compute to perform graph classification, we test with kernel regression and $C$-SVM as the final classifier. In our experiments, the regularization parameter $C$ in $C$-SVM is determined using grid search from 120 values evenly chosen from $[10^{-2}, 10^4]$, in log scale.

We would like to remark that GNTK has strictly smaller number of hyper-parameters than GNN since we do not need to tune the learning rate, momentum, weight decay, batch size and the width of the MLP layers for GNTK. Furthermore, we find on bioinformatics datasets, we get consistently good results by setting the number of BLOCK operations to be 10, the number of MLP layers to be 1 and $c_u$ to be $1/|\mathcal{N}(u)|$. We get 75.3% accuracy on PROTEINS, 67.9% on PTC, and 83.6% on NCI1. For social network datasets, by setting the number of BLOCK operations to be 2, the number of MLP layers to be 2 and $c_u$ to be 1, we get 76.7% accuracy on IMDB-B, 52.8% on IMDB-M, and 83.3% on COLLAB.