[Reviews · NeurIPS 2019]

Reviewer 1



Pros. - The derivation of explicit formulae, which I think is correct, is neat and requires work. - The sample complexity analysis requires some work in math. Cons. - Only BLOCK and READOUT layers are analyzed. Since the title is GNTK, it may be better to include analysis of graph convolutional layers. - The sample complexity analysis is a bit rough. Theorems 4.2 and 4.3 are more like technical lemmas whose usefulnesses are in doubt per se. Furthermore, the argument of polynomial’’ sample complexity relies on the condition given in line 193 (please consider numbering the equations). There is no discussion in relation to the applicability of this condition. - The experiments showed in general GNTK matches the performance of the state of the art. As the method of NTK is still quite new, it is okay for me that the performances were not strictly better. But the discussion on the relation between GNTK and GNN (subsections 5.2) is too coarse; figures 2 is simply restating what the community has already known; I am not sure what the authors wished to illustrate by figure 3. There were surely many things interesting going on, but I think unfortunately the authors failed to give a convincing account of them. Minor points: - It would be nice if equations are numbered all the time; - Sometimes the use of the Big-Oh notation is not accurate - one cannot say something $\le O(\cdot)$; instead, please say something $= O(\cdot)$. Questions. 1. Could the authors give a short derivation outline of Theorem 4.1 based on (Bartlett and Mendelson, 2002)? I am sorry that I am not very familiar with the result used. 2. Can the authors empirically show the magnitude of different parameters in the RHS of line 357, and if the conditions on line 193 are met? 3. I am not sure about the conclusion of polynomial sample complexity even when line 193 is satisfied. It seems possible to choose the coefficients $\alpha_{2l}$ and $\boldsymbol{\beta_{2l}}$ in a way that the risk bound is no longer polynomial in $n$. Besides, how tight is this bound? Conclusions. In general, this well-written paper did provide the community with explicit formulae of NTK for GNNs composed of BLOCK and READOUT layers, but the contribution is somewhat incremental. Besides, it is not immediately clear how good the sample complexity analysis is in real settings. Although a handful of experiments were performed, the interpretation of the results is lacking and sometimes too weak. -------------------------- In the rebuttal, the authors have addressed some of my questions and concerns in a satisfactory manner. Considering the significant novelty this submission provides, I decide to raise my score to "Weak Accept". If possible, I hope the authors can address the following questions in the revised manuscript: -- Regarding the new figures showing testing loss against the training sample size, how to interpret them is a subtle question since, as the authors have also pointed out, the theoretical bound in the paper is not tight. I wish to see more details on how these simulations empirically showed the applicability of their results. -- I now understand that Figures 1 and 2 in the original submission were used to show that GNTKs have similar performance to GNNs. But I disagree with the authors on whether the figure 3 shows jumping connections are effective as I cannot tell the difference in figure 3. As Reviewer_2 has suggested, a statistical test would greatly enhance the credibility of this claim, which is lacking now. That being said, I still feel that the discussions on the theoretical results can be further improved such as on the conditions on line 193, particularly the assumptions made in Theorem 4.2, I am still not sure how in general that assumption would hold.

Reviewer 2



The paper proposes a way to derive graph kernels from Graph Neural Architectures, building on similar results on Neural Tangent Kenrels. the proposed formulation allows for a deeper analysis compared to GNNs: the paper proposes a sample complexity analysis on functions of a certain form. page 3, section 3.1 line 118. If I'm not mistaken, the \theta inside the f function in the definition of u(t) should depend on t. Otherwise, u(t) does not depend on t, and it looks weird. Experimental results: results are not astonishing. The proposed method works better than competitors on COLLAB dataset. For other datasets, the variance is pretty high and the results are within a standard deviation from baselines. A statistical test may help in interpreting the experimental results. Also saying that the proposed method is always comparable with the neural counterpart may be an interesting result. On the other hand, the proposed method remains always comparable to GNNs (usually performing slightly better), that is a good thing. Page 8, line 243: you state that Figure 3 shows that jumping knowledge improves the performance of GNTK in both IMDB and NCI Actually, it looks like for IMDB Jumping knowledge doesn't change the results much, while for NCI1 it slightly improves the performance of the best GNTK architecture, but we're probably speaking of 0.002 points in accuracy at most (extrapolating from the plot). Computational complexity: experimental results would have benefitted from a computational complexity analysis / running time comparison between the different methods. Usually Graph Kernels + SVM are faster to compute compared to GNNs, that require GPUs to run efficiently. One of the main points of the present paper may be of providing kernels with generally comparable performance compared to GNN, but with a faster implementation. However, I could't find any reference on computational complexity in the paper. Authors state in the reproducibility list that the release of source code and of the adopted datasets is not applicable. I don't agree with that, however authors did include the source code and the datasets as supplementary material. Thus, I assume is in authors' intention to release the code if the paper gets accepted. Minor: page 1 line 27: "worse practical performance" : compared to what? page 7 line 225: IMDBBINARY -> IMDB-B Figure 2 is not readable in black and white printing Conclusions are missing ####AFTER REBUTTAL I acknowledge that authors responded in a satisfactory way to my comments.

Reviewer 3



Originality: The idea is very inspiring in the paper. Motivated by GNN and over-parameterized neural networks, a Graph Neural Tangent Kernel is presented. Quality: The presented algorithm seems technical sound. The experimental results are good. My concern is that there are too many parameters to search in such small datasets, make it very difficult to use in practice. Clarity: The paper is well written. The authors have nicely introduced the algorithm and the theory behind that. I like figure 1 which shows how to transform the GNN to GNTK. Significant: Currently, theoretical studies for GNN is rather limited in the research community. This paper would enhance theoretical studies in this area.

[Author Response · NeurIPS 2019]

We thank all reviewers for their valuable feedback and appreciating our technical contributions. We first address some
common concerns.

-*More discussion on experimental results*:

• First, we discuss more about the relation between GNTKs and GNNs. In Section 5.2, through ablation studies, we
aimed to verify 1) whether performance of GNNs is similar to GNTKs and 2) whether various techniques used for
improving performance of GNNs are transferable to GNTKs. For example, in Figure 2, while the observation that
GNNs with more layers perform better on bioinformatics data is not new, we observed the same trend for GNTKs. In
Figure 3, we also show that jumping knowledge, which has been shown to be effective in GNNs, is also effective in
GNTKs. We will elaborate more on these connections in the final version.

• Second, Reviewer #2 raised a great point about computational complexity. Indeed, running GNTK is much faster
than running GNN. On IMDB-B dataset, running GIN with the default setup (official implementation of [26]) takes 19
minutes on a TITAN X GPU and running GNTK only takes 2 minutes. We will add more details.

-*Reproducibility*: We will open-source our code and datasets. We mistakenly chose "not applicable" for this question in
Reproducibility Response.

**To Reviewer #1:**

-*Graph convolutional layer*: Our paper does contain a graph (spatial) convolutional layer which we call the aggregation
step ($\sum_{v \in \mathcal{N}(u) \cup \{u\}} \mathbf{h}_v^{(\ell-1)}$). See line 89, 96 for GNNs and line 148 for GNTKs. We will clarify this more explicitly.

-*Lemma 4.2/4.3, conditions on line 193*: The aim of our theoretical analysis is to characterize the function class which
can be efficiently learned by GNTK. This is a standard type of results in learning theory. The sample complexity
depends on $\alpha$, the coefficient of polynomials, and the norm of $\beta$. This kind of dependency is standard in learning
polynomials and functions that can be expressed as summation of polynomials with fast decaying coefficients. If $\alpha$
or the norm of $\beta$ is large, we do need more samples. See [1] and references therein. The tightness of the bound is an
open problem. In general, it is impossible to verify these conditions on real datasets, as this is a quantity related to the
underlying function which we cannot observe. Nevertheless, empirically we use simulations to show the applicability
of these theoretical results, as given in Fig. 1. We will include these simulation results in the paper.

(a) $\ell = 1$.   (b) $\ell = 2$.   (c) $\ell = 4$.

Figure 1: We perform simulations using 1000 graphs in the IMDB-B dataset. For each graph, we generate the label according
to $y = \sum_{u \in V} (\bar{h}_u^\top \beta)^\ell$ for $\ell \in \{1, 2, 4\}$, where we sample $\beta$ from the uniform distribution over the unit ball (cf. Eq. (4) in our
submission). By construction, these functions satisfy the condition in line 193. We use 100 samples for testing and use 100, 300,
500, 700 or 900 remaining samples for training. We repeat each simulation for 100 times and report the testing loss.

-*Theorem 4.1*: This is a classical result for kernel regression. The high-level idea is to use Rademacher complexity to
bound the difference between empirical loss and population loss, and use $\mathbf{y}^\top \mathbf{\Theta}^{-1} \mathbf{y} \cdot \mathrm{tr}(\mathbf{\Theta})$ to bound the Rademacher
complexity. The whole proof is given in [3] and we will provide a short outline in the final version.

**To Reviewer #2:**

-*Statistical test:* We will add a statistical test in the final version. Thanks for the suggestion.

-*Line 118 / typos / missing conclusions*: We will fix them accordingly. Thanks for pointing out.

**To Reviewer #3**:

-*Eq. (2)*: For the BLOCK operation with $R = 2$, we first apply the aggregation operation to gather local information,
and then apply a two-layer MLP to create non-linearity. We will clarify this in the final version.

-*Experimental details:* For experiments, we use the same setup as in [26]. We have provided some descriptions of the
experimental setup in Section B. We will add more detailed description in the final version.

-*Too many parameters*: GNTK do have some hyper-parameters. However, note that GNTK has strictly smaller number
of hyper-parameters than GNN since we do not need to tune the learning rate, momentum, weight decay, batch size and
the width of the MLP layers for GNTK. Furthermore, we found on bioinformatics datasets, we got consistently good
results by setting the number of BLOCK operations to be 10, the number of MLP layers to be 1 and $c_u$ to be $1/|\mathcal{N}(u)|$.
We get 75.3% accuracy on PROTEINS, 67.9% on PTC, 83.6% on NCI1. For social network datasets, by setting the
number of BLOCK operations to be 2, the number of MLP layers to be 2 and $c_u$ to be 1, we get 76.7% accuracy on
IMDB-B, 52.8% on IMDB-M, and 83.3% on COLLAB. We will discuss this point in the final version.

[Meta-Review · NeurIPS 2019]

As the reviewers' concerns were clarified in the author feedback and discussion, they unanimously recommend to accept the paper.